# Trabecular Bone Score (TBS) in Individuals with Type 2 Diabetes Mellitus: An Updated Review

**DOI:** 10.3390/jcm12237399

**Published:** 2023-11-29

**Authors:** Alexandra-Ioana Trandafir, Oana-Claudia Sima, Ana-Maria Gheorghe, Adrian Ciuche, Anca-Pati Cucu, Claudiu Nistor, Mara Carsote

**Affiliations:** 1PhD Doctoral School, Carol Davila University of Medicine and Pharmacy, 050474 Bucharest, Romania; alexandratrandafir26@gmail.com (A.-I.T.); oanaclaudia1@yahoo.com (O.-C.S.); anamaria.gheorghe96@yahoo.com (A.-M.G.); ancutapati@gmail.com (A.-P.C.); 2Department 4—Cardio-Thoracic Pathology, Thoracic Surgery II Discipline, Carol Davila University of Medicine and Pharmacy, 050474 Bucharest, Romania; adrianciuche@gmail.com; 3Thoracic Surgery Department, Dr. Carol Davila Central Military Emergency University Hospital, 010242 Bucharest, Romania; 4Department of Endocrinology, Carol Davila University of Medicine and Pharmacy, 050474 Bucharest, Romania; carsote_m@hotmail.com; 5Department of Clinical Endocrinology V, C.I. Parhon National Institute of Endocrinology, 020021 Bucharest, Romania

**Keywords:** trabecular bone score, TBS, bone microarchitecture, bone mineral density, diabetes mellitus type 2, diabetic bone disease, DXA, fracture, osteoporosis, metabolic bone disease

## Abstract

Bone fragility is a complication of type 2 diabetes mellitus (T2DM) that has been identified in recent decades. Trabecular bone score (TBS) appears to be more accurate than bone mineral density (BMD) in diabetic bone disease, particularly in menopausal women with T2DM, to independently capture the fracture risk. Our purpose was to provide the most recent overview on TBS-associated clinical data in T2DM. The core of this narrative review is based on original studies (PubMed-indexed journals, full-length, English articles). The sample-based analysis (n = 11, N = 4653) confirmed the use of TBS in T2DM particularly in females (females/males ratio of 1.9), with ages varying between 35 and 91 (mean 65.34) years. With concern to the study design, apart from the transversal studies, two others were prospective, while another two were case-control. These early-post-pandemic data included studies of various sample sizes, such as: males and females (N of 245, 361, 511, and 2294), only women (N of 80, 96, 104, 243, 493, and 887), and only men (N = 169). Overall, this 21-month study on published data confirmed the prior profile of BMD-TBS in T2DM, while the issue of whether checking the fracture risk is mandatory in adults with uncontrolled T2DM remains to be proven or whether, on the other hand, a reduced TBS might function as a surrogate marker of complicated/uncontrolled T2DM. The interventional approach with bisphosphonates for treating T2DM-associated osteoporosis remains a standard one (n = 2). One control study on 4 mg zoledronic acid showed after 1 year a statistically significant increase of lumbar BMD in both diabetic and non-diabetic groups (+3.6%, *p* = 0.01 and +6.2%, *p* = 0.01, respectively). Further studies will pinpoint additive benefits on glucose status of anti-osteoporotic drugs or will confirm if certain glucose-lowering regimes are supplementarily beneficial for fracture risk reduction. The novelty of this literature research: these insights showed once again that the patients with T2DM often have a lower TBS than those without diabetes or with normal glucose levels. Therefore, the decline in TBS may reflect an early stage of bone health impairment in T2DM. The novelty of the TBS as a handy, non-invasive method that proved to be an index of bone microarchitecture confirms its practicality as an easily applicable tool for assessing bone fragility in T2DM.

## 1. Introduction

Bone fragility is a complication of type 2 diabetes mellitus (T2DM) that has been identified in recent decades as one of many others of this complex disorder. While bone mineral density (BMD) measured by Dual-Energy X-ray Absorptiometry (DXA) is considered the gold standard for evaluating osteoporosis, BMD-DXA does not reflect the microarchitecture changes that are also a part of the osteoporosis-associated panel. On the other hand, trabecular bone score (TBS) is obtained from DXA scans of the lumbar spine being strongly associated with bone (trabecular) microarchitecture and providing information independently of BMD. TBS appears to be more accurate than BMD of the lumbar spine in diabetic bone disease, particularly in menopausal women diagnosed with T2DM in order to predict fracture risk [1,2,3]. Thus, assessing the impairment of bone status in T2DM is mandatory.

T2DM, a chronic metabolic condition, has a massive public health burden coming from its epidemiologic impact and the large panel of co-morbidities. The global prevalence of T2DM was estimated to be 10.5% in 2021, being expected to rise to 12.2% in 2045 [1]. T2DM is associated with the development of a variety of complications, including retinopathy, nephropathy, neuropathy, and cardiovascular diseases [2]. Recent studies added bone fragility to this list, namely diabetic bone disease or diabetes-induced osteoporosis, with increased fracture risk in association with muscle-associated anomalies, particularly diabetes-related sarcopenia [3,4,5,6]. Simultaneous T2DM and osteoporosis lead to worse health outcomes, thus their adequate recognition and management are important [7]. Patients with T2DM have a 19% increased risk of any fragility fracture and higher post-fracture mortality than the non-diabetic population [8,9]. Therefore, the increasing number of diabetic patients requires a detailed evaluation, including an evaluation of fracture risk.

Conventionally, BMD and associated T-score according to central DXA are recommended for screening in order to evaluate the fracture risk; however, BMD does not fully reflect fracture risk because the material properties of bone tissue also play a role in the ability of the whole bone to resist against a stress fracture. It cannot distinguish between cortical and trabecular bone, and it offers limited information on bone quality [10,11]. Bone fragility in diabetes results not only from alterations in bone mineralization, but also from alterations in bone microarchitecture, which can be evaluated via TBS [12]. TBS is indirectly calculated from using experimental variograms of two-dimensional gray-level projection images obtained from DXA scans [13,14]. TBS indicates a reduced number of trabeculae and less connectivity, as well as trabecular separation (overall, impaired bone microarchitecture) [11]. There are other imaging modalities for assessing bone quality, including high-resolution peripheral quantitative computed tomography (HR-pQCT) and high-resolution magnetic resonance imaging (HR-MRI). Among these methods, TBS is the most widely used instrument to study bone microarchitecture in diabetes because it is largely available, associating low costs and reduced radiation (based on X-ray according to a standard lumbar DXA scan). Currently, TBS represents the most practical tool for clinicians to address the bone status in terms of microarchitecture, and this evaluation is not limited to the specialists in the field of osteoporosis, being an easy-to-use tool upon software applications on a DXA device [15]. Consequently, TBS remains an important tool to be analyzed in relationship with T2DM.

Generally, T2DM is characterized by normal or high BMD, but also an increased risk of fragility fractures, an apparent paradox suggesting that independent factors other than BMD may influence the fracture risk [16]. Recent studies have indicated that lower values of TBS were associated with an increased risk of fragility fracture, and, importantly, the association was independent of BMD. Overall, the affected bone microarchitecture is recognized as a major factor contributing to the risk of fracture in T2DM, diabetic bone disease representing a major co-morbidity as part of an already complicated panel, [17,18,19], noting once again the role of understanding and studding the bone fragility in T2DM due to the increased burden in terms of morbidity and even mortality especially when it comes to the hip fractures that affect a large number of the population, hence the importance of addressing such topic in the modern era.

### 1.1. The Scientific Background of Why the Study Topic Must Be Conducted

The scientific rationale of detecting and improving the strategy of addressing the osteoporotic fractures in the general population (and in type 2 diabetic individuals in particular) comes from a major burden that may be reflected by impressive data: the mortality within the first year following a hip fracture may rise to 25% while disability and dysfunctionality might affect up to 60% of affected persons following this type of fracture; the projections of osteoporotic fractures as increasing numbers of patients involve both females and males; a call for action has been released by World Health Organization to address the issue of fragility fractures and osteoporosis detection due to the associated burden; remaining lifetime probability of suffering from a fracture is one out of three females and one out of five men aged over 50 in Europe and, respectively, one out of two and one out of four in the US; the number and costs of hospitalizations due to fractures surpasses those related to cardiovascular events; the panel of osteoporosis-associated disability represents a heterogeneous spectrum that is also reflected by the aggravation of other co-morbidities, increased costs, and an overall impaired quality of life. On the other hand, only 2 out of 10 patients with osteoporosis are recognized with this diagnosis, and only 3 out of 10 subjects with an osteoporotic fracture who should receive specific anti-osteoporotic medication are actually treated for this condition in order to reduce the increased fracture risk, while almost half of the patients that start this medication will no longer be compliant after the first year since drug initiation [20,21,22,23,24,25,26].

Importantly, the burden of T2DM comes not only by its alarming globally increasing incidence, including in younger people and low-income countries (affecting from 6% to 22% of the entire population at different age segments) but, also, by the associated morbidity and mortality; for instance, it occupies the ninth place among the most frequent causes of death and the seventh place among the conditions with the most impactful quality of life in terms of DALYs (disability-adjusted life year), equally affecting females and males, with the costs per capita of diabetic care being three to nine times (in cases of multiple complications, including bone fragility) higher than the mean healthcare expenditure per capita in non-diabetic populations [27,28,29].

While the modern era is marked by an aging population [30], both osteoporosis and T2DM might come as mandatory issues to be addressed from all multidisciplinary perspectives, from primary to tertiary care, from medical to social approaches [31,32,33,34,35,36,37,38,39]. Noting these important mentioned aspects, the assessment of bone fragility among T2DM comes as urgent.

### 1.2. Aim

Our purpose was to provide a most recent overview on TBS-associated clinical data in T2DM. We followed several aspects such as TBS values in relationship with DXA-BMD, the reflection of glucose profile into TBS, associated anomalies of metabolism that might interfere with bone status in these patients, the correlation between TBS and bone turnover as reflected by blood/urinary biomarker assays and the specific intervention against osteoporosis in T2DM with potential effects on TBS improvement. The novelty of such a literature-based update should provide new perspectives on the means to assess the bone fragility status among T2DM patients in relationship with TBS that is placed in the core of these assessments; its correlation with traditional bone evaluation based on DXA, particularly BMD and BMD-derived T-score, with blood and/or urinary bone turnover markers and, also, with the panel of T2DM such as glucose profile (fasting glucose, postprandial glycaemia, and glycated hemoglobin A1c) and specific therapy against diabetes.

## 2. Methods

Based on the hypothesis that diabetes is associated with abnormal bone microarchitecture, T2DM correlates with a lower TBS value when compared to non-diabetic subjects. We therefore set out to search the literature for studies that have examined TBS in patients with T2DM published in PubMed (full-length articles, published in English, between 1 January 2022 and 30 September 2023), and we introduced our analysis according to a narrative review. The keywords used for the search were “trabecular bone score” (alternatively, “TBS”) and “diabetes”.

Inclusion criteria are represented by original studies of any study design conducted in adults (females and males) diagnosed with T2DM. We excluded: review papers, case reports and case series, editorials, letters to the editor, editorials, conference abstracts/reports, animal/experimental studies, data on type 1 diabetes mellitus (T1DM) and other secondary causes of diabetes, as well prediabetes/impaired glucose tolerance, and pediatric data. This 21-month sample-based analysis followed several micro-sections such as DXA-based BMD and TBS aspects, TBS values in relationship with glucose control, bone turnover markers, and interventional approaches for diabetes-associated osteoporosis.

## 3. TBS as Practical Tool to Assess the Bone Health of Adults with T2DM

### 3.1. Sample-Based Study Regarding TBS in Diabetic Bone Disease

Out of 74 papers, we identified 11 original studies that examined the association between T2DM and TBS. A total of 4653 patients were included, with a higher proportion of females (female/male ratio of 3054/1599), aged between 35 and 91 years (with a mean age of 65.34 years) (Figure 1).

The association between diabetic bone disease and TBS was found in women and to a lesser extent in men, thus confirming prior data (specifically post-menopausal females). The core findings were a confirmation of prior data, namely a lower TBS in diabetic than non-diabetic controls [40,41,42,43,44,45,46,47,48,49,50] (Table 1).

### 3.2. Lower TBS and Higher BMD in T2DM

Most of the 11 studies confirmed that the patients with T2DM associated a normal or increased BMD, but lower bone quality as measured by lumbar DXA-based TBS. Generally, BMD is positively correlated with body mass index, which in subjects with metabolic issues, including T2DM, may be abnormally high. On the other hand, TBS reflects a damage of bone quality that is situated in relationship with glucose profile anomalies in diabetic populations, TBS being considered a better fracture predictor than DXA-BMD in T2DM. Thus, BMD may underestimate the risk of fracture in T2DM; generally, TBS is a complementary tool to BMD since the magnitude of fracture prediction is independent [51,52].

We highlighted three studies within our 21-month sample-based analysis that particularly reflected T2DM’s lowering effect on TBS and increased BMD [44,47,48]. A cohort study (N = 511) conducted by Haeri et al. [44] included both females with T2DM of 80 years and older (N = 433, mean age of 80.6 ± 8.0 years) and males (N = 78, mean age of 82.4 ± 8.3 years). Diabetic women had a higher lumbar spine BMD (1.106 ± 0.200 *versus* 1.017 ± 0.204 g/cm^2^, *p* = 0.0003), femoral neck BMD (0.695 ± 0.123 *versus* 0.651 ± 0.114 g/cm^2^, *p* = 0.0463), but similar total hip BMD (0.780 ± 0.331 *versus* 0.734 ± 0.235 g/cm^2^, *p* = 0.6255) compared to females without T2DM. TBS was statistically significantly lower in the T2DM group *versus* the control group represented by 328 non-diabetic females (1.211 ± 0.172 *versus* 1.266 ± 0.136, *p* = 0.0299). However, these differences were not consistent within the male group (which was of a very small sample size), whereas no difference was observed between lumbar spine BMD (*p* = 0.6851), femoral neck BMD (*p* = 0.5559), total hip BMD (*p* = 0.4848), and TBS (*p* = 0.7935), respectively, *versus* controls (N = 60 non-diabetic males). Of note, all the patients were senior residents at long-term care facilities (more than a half of them were affected by pre-frailty or frailty), being among the very few studies on TBS in this distinct population subgroup [44]. Collaterally, we mention the results of the FRODOS cohort (from 2018) that included 2257 menopausal females with an average TBS of 1.203 ± 0.121 (a value ≤ 1230 indicating a degraded bone microarchitecture was identified in 55.3% of them): a history of fragility fracture due to frailty was among the factors that were associated with decreased TBS (according to a multiple linear regression), also, including patients’ age, anthropometric elements such as weight and height, lumbar T-score, glucocorticoid exposure, and T2DM [53]. Alternatively, another type of approaching TBS results in relationship with a frailty component includes hemodialysis-associated frailty in subjects suffering from kidney disease mineral and bone disorder. We mention a prior observational study on 47 patients (40.4% had DM) that showed that TBS (TBS iNsight) was statistically significantly reduced in the subgroup with a history of cardiovascular events (*p* < 0.001) *versus* the cohort without these complications, TBS being associated with an increased age. On the other hand, lower TBS values were correlated with a higher mortality, and, also, as seen in a non-frail population, with an increased fracture incidence [54].

Another study was consistent with previous studies; specifically, Ubago-Guisado et al. [48] revealed that patients with T2DM (N = 111) presented a statistically significant increased BMD compared with the control group (N = 134 non-diabetic individuals) at the level of lumbar spine (1.043 ± 0.198 *versus* 0.964 ± 0.167 g/cm^2^, *p* = 0.001), total hip (1.052 ± 0.175 *versus* 0.921 ± 0.166 g/cm^2^, *p* < 0.001), as well as femoral neck (0.822 ± 0.159 *versus* 0.748 ± 0.128 g/cm^2^, *p* < 0.001). As expected, TBS was statistically significantly lower in the T2DM group compared with the control group (1.074 ± 0.187 *versus* 1.291 ± 0.110 g/cm^2^, *p* < 0.001). In this study, both conventional DXA and 3D-DXA confirmed BMD results (higher values in diabetic *versus* non-diabetic individuals) and, also, confirmed a positive correlation between BMD and body mass index; overall, this clinical parameter is a positive predictor of areal BMD, but a negative one of TBS [48].

Similarly, Dule et al. [47] showed that females with T2DM (N = 126 *versus* 117 non-diabetic, age-matched controls) have impaired bone microstructure as assessed by TBS, although they have normal or increased BMD. T2DM was associated with low TBS according to an adjustment for age, menopausal status, and body mass index (OR = 2.47, 95% CI: 1.19–5.16, *p* = 0.016). Conversely, diabetes status was confirmed to be correlated with higher lumbar BMD (OR = 0.43, 95% CI: 0.21–0.89, *p* = 0.024), but not with BMD values measured in the other central DXA sites. Moreover, the ratio of subjects with normal DXA was lower in diabetic females group (5%) than controls (12%, *p* = 0.04) [47].

On the contrary, another study, according to our methods, confirmed a TBS reduction in T2DM, but not a BMD increase at central DXA [50] or higher BMD, but not decreased TBS in diabetic *versus* non-diabetic females (a lower TBS was, however, confirmed in T2DM males) [43], while two other studies showed similar BMD and TBS values in diabetic *versus* non-diabetic adult males [41] or women [45]. For instance, Kim et al. [45] did not find statistically significant differences at baseline BMD-DXA evaluation between the diabetic (N = 49 postmenopausal females, with mean age of 73 years) and non-diabetic patients (N = 55 non-diabetic controls, with mean age of 66 years) regardless of the DXA site (lumbar spine: *p* = 0.098; femoral neck: *p* = 0.663; total hip: *p* = 0.607), neither with concern to TBS values (*p* = 0.294). Of note, this was a small sample size study [45]. A cross-sectional study by Naseri et al. [50] revealed that TBS was statistically significantly decreased (1.280 ± 0.111 *versus* 1.343 ± 0.101, *p* < 0.001) in diabetic menopausal subjects (N = 348; mean age of 61.40 ± 7.93 years) than in non-diabetic controls (N = 539; mean age of 55.13 ± 6.61 years), but there were no differences in BMD at the lumbar spine (*p* = 0.904), femoral neck (*p* = 0.402), and hip (*p* = 0.108) between the two groups after adjusting for age and body mass index [50]. Yet, BMD changes were identified at the level of distal radius and total forearm (with decreased values in the diabetic group *versus* controls, *p* < 0.05) [50], which emphasize that central DXA sites might not be useful discriminators among T2DM females with respect to diabetic bone disease and associated fracture risk assessment; and, of course, a normal DXA in one T2DM menopausal lady does not mean that the patient is not at higher fracture risk, thus she might become a candidate for anti-osteoporosis medication while the role of DXA-BMD as a marker of medication response during follow-up remains questionable.

The majority of the data agree that the mostly (menopausal) female population reflects the TBS decrease amid T2DM; we identified one study that did not confirm this aspect in women, only in men. This is the Bushehr Elderly Health program that enrolled T2DM subjects and those who received glucocorticoids, a total of 2294 individuals (females/males ratio of 1182/1112, with a mean age of 69.3 ± 6.3 years); the cohort was divided into patients confirmed with T2DM (N = 726) and non-diabetic (N = 1568). The T2DM group had a higher lumbar BMD in both sexes (females: *p* < 0.001; males: *p* < 0.001) in comparison with non-diabetics, whilst TBS was reduced only in male groups (*p* = 0.03). Notably, body mass index was the most important factor of influence for BMD-DXA, as well as TBS [43].

### 3.3. Glucose Profile and Diabetes Characteristics: TBS Changes

Weak glycemic control in T2DM has been correlated with a lower TBS, but not all authors agree. A heterogeneous spectrum of results has been identified so far with regard to diabetes duration and control, as well as specific medication against diabetes and their impact on bone microarchitecture. As expected, good long-term disease compensation, as reflected by the values of glycated hemoglobin A1c, fasting glucose level, and postprandial glycaemia, are correlated with higher TBS values, but not necessarily with a tide BMD association [55,56,57]. While an optimum glucose status is generally required to prevent any type of T2DM complications, the exact target values of these mentioned parameters with respect to the most beneficial results on bone health are still an open issue [56,57,58].

We identified three studies that addressed the issue of correlating diabetes control to TBS values, either in terms of glycated hemoglobin A1c levels or T2DM-related complications such as microvascular disease [40,41,46]. Particularly, two out of these three studies provided longitudinal data with respect to a 12-month mean of hemoglobin A1c [40] or an analysis of diabetic subgroups based on a T2DM duration with a cut-off of 5 years [46]. With regard to the studied population, one cohort included both males and females [40], while the others included only men [41] or only women [46]. As mentioned, most of the previous studies on T2DM profile (including our sample-based analysis) were conducted in females, particularly postmenopausal women, as expected according to the general recommendations starting with 2015 since the ISCD (International Society for Clinical Densitometry) Official Position was released regarding TBS applications [59], following the first years of clinical studies on TBS [60,61,62,63,64].

A longer duration of T2DM and higher HbA1c levels in menopausal females were associated with increased hip BMD (*p* = 0.001) and decreased TBS (*p* = 0.02) according to a cross-sectional study on 493 females of 65 years old or older [46]. Prior data showed a higher BMD and lower TBS in uncontrolled diabetes *versus* normo-glycemic T2DM patients [65], but variable results have been attained depending on studied population with respect to a potential tide connection between T2DM control and skeleton impairment [66,67]. Also, Ballato et al. [41] studied 169 diabetic men, aged between 35 and 65 years (an average of 51.4 ± 7.5 years, disease duration of 7.75 ± 6.3 years) who were divided according to glycated hemoglobin A1c levels (a mean value for the prior year with a cut off of 7%) into those with good (≤7%) *versus* poor glycemic control (<7%); both groups were compared with younger patients without T2DM. At baseline, no differences with respect to BMD at the lumbar spine (*p* = 0.87), femoral neck (*p* = 0.12), total hip (*p* = 0.43), and TBS (*p* = 0.28) among these groups were identified. However, other indices of bone strength such as tibial stiffness and failure load were lower within the group with uncontrolled diabetes *versus* healthy controls (*p* = 0.01, respectively, *p* = 0.009) [41].

Moreover, diabetic microvascular disease may be correlated with a reduced TBS value, as an additional clue of poor glycemic control. Maamar El Asri et al. [40] published a cohort that examined the relationship between this complication (in terms of nephropathy, neuropathy, and/or retinopathy) and TBS in patients with T2DM: males who were ≥50 years (N = 177) and postmenopausal females (N = 184), with a mean age of 63.8 ± 9.7 years (age range between 47 and 91 years). The subjects experiencing this microvascular co-morbidity (TBS of 1.252; 95% CI: 1.230−1.274) had a statistically significant lower TBS than those without it (1.281; 95% CI: 1.267−1.295) after adjusting for the confounding variables (*p* = 0.034). In a similar way, subjects with diabetic microvascular disease had an elevated level of glycated hemoglobin A1c accompanied by TBS < 1.230, regardless of the disease duration. In contrast, BMD showed no differences between the two groups regardless of the site: lumbar spine (*p* = 0.70), femoral neck (*p* = 0.42), and total hip (*p* = 0.19) [40]. Notably, we looked at similar studies on PubMed with no timeline restriction and found no other analysis to specifically address microvascular disease and TBS; only one study showed a reduced bone quality in terms of low TBS and decreased BMD in patients with microvascular damage due to systemic sclerosis when compared to healthy controls [68].

### 3.4. Metabolic Components-Associated Impact on Bone Quality

Whole-body DXA may provide the body composition parameters in terms of android or gynoid fat mass (and, also, android-to-gynoid fat ratio), both being correlated to BMD [69]. Fat distribution is important for bone health; a positive association between body mass index and TBS has been found, and gynoid fat turned out to be a protective factor and android fat was a risk factor for TBS reduction. Also, most authors agree that TBS precision error is not influenced by fat mass [70]. Dramatic weight loss, as seen after bariatric/metabolic surgery, may induce TBS damage in 25% of the individuals, and one of the mechanisms involves post-operatory short- and long-term abnormal fat–bone–muscle–gastrointestinal tract cross-talks [71].

With regard to the metabolic components’ influence on TBS, we mention two studies: one addressed the issue of fat mass [42] and the other analyzed multiple metabolic components, particularly serum lipids profile [47]. Menopausal females aged between 50 and 75 years with reduced TBS had lower gynoid fat mass and higher android/gynoid fat mass ratio (*p* = 0.004, and *p* < 0.0001, respectively) according to one non-interventional, cross-sectional study; 96 females with T2DM (with a median T2DM duration of 15 years) who had normal BMD were included (mean age of 64 years old and an average time since menopause of 16 years). This comparative study revealed that a significant ratio of postmenopausal females with T2DM (in association with normal BMD) may have impaired bone microarchitecture as reflected by a decrease in TBS in 44.8% of the subjects, thus being at a higher fracture risk that is not captured by traditional DXA-BMD, as we previously specified. Also, 11.3% of them with a value of TBS > 1.31 and 32.6% of them having a TBS ≤ 1.31 had at least one prevalent osteoporotic (low trauma) fracture (*p* = 0.02). Subjects with a TBS ≤ 1.31 had lower body mass index *versus* those with TBS > 1.31 (*p* = 0.007). However, the subgroups with a TBS cutoff of 1.31 (≤ or >) were similar with respect to glycated hemoglobin A1c, PTH (parathormone), 25-hydroxyvitamin D, bone formation marker alkaline phosphatase, serum levels of calcium and phosphorus [42].

Also, Dule et al. [47] identified in T2DM females a positive correlation between TBS and HDL (high-density lipoprotein) cholesterol (*p* = 0.029) and serum vitamin D (*p* = 0.017), respectively; the multivariate regression model showed that plasma HDL-cholesterol represents the most important predictor of TBS, irrespective of patients’ age and menopausal status, clinical elements of potential metabolic components such as body mass index and waist circumference, cholesterol-lowering drugs exposure (statins), the level of physical activity, and serum 25-hydroxyvitamin D [47].

### 3.5. Bone Turnover Markers in Diabetes-Induced Osteoporosis

Diabetic bone disease includes, among other things, a blunt bone turnover markers profile, which is not unanimously found to be correlated with TBS according to transversal and longitudinal cohorts, in both treatment-naïve patients and subjects who were under osteoporotic drugs. Generally, these blood and urinary biomarkers have large inter- and intra-individual variations, thus their behavior is hardly predictable in one subject, and rather they are useful in clinical studies, particularly to serve as surrogates during follow-up for fracture risk reduction under specific anti-osteoporotic intervention [57,72,73,74,75].

We mention the results of three studies on TBS and T2DM that provided additional information on these markers [40,41,48]; another one included the markers’ profile as part of anti-osteoporosis medication-associated follow-up [45]. The study of Ubago-Guisado et al. [48] addressed a large panel of metabolic elements; among others, serum P1NP as a bone formation marker represented a negative predictor of TBS values (*p* ≤ 0.01) as opposite to the bone resorption marker CTX (*p* = 0.02) [48]. Also, the study of Ballato et al. [41] showed that diabetic men with poor glycemic control as reflected by the glucose profile during the year prior had statistically significant lower osteocalcin and CTx levels *versus* non-diabetic controls (*p* < 0.001 for both markers) [41]. The transversal study of Maamar El Asri et al. [40] (N = 361) found no differences with respect to P1NP as well as β-CTX between the individuals with microvascular disease and those without this diabetic complication (as well as similar BMD, but lower TBS). Notably, 25-hydroxy-vitamin D was lower in the group with the mentioned complication (8.3 ng/mL *versus* 21.6 ng/mL; *p* = 0.0001) [40].

### 3.6. Interventional Approach for Osteoporosis in T2DM Patients

The role of TBS as an everyday tool that is meant to follow-up the effects of anti-osteoporotic drugs, as seen for DXA-BMD/T-score, is still controversial, but numerous studies addressed this issue; whether TBS is more useful in T2DM patients than BMD is yet to be proven. On the other hand, so far, no ideal drug against diabetes-induced osteoporosis has been found, and typically the management of the diabetic osteoporotic patients is similar to the non-diabetic individuals in addition to a good glycemic control [6,76,77,78].

Our 21-month sample-based analysis identified two clinical studies on bisphosphonates with intravenous administration in T2DM patients [45,49]. A prospective, interventional, open-label trial (between 2018 and 2020) was conducted on diabetic and non-diabetic groups that received 150 mg monthly ibandronate for 12 months [45]. The results showed a similar BMD increase in both cohorts at each central DXA site, with similar blood–bone turnover markers profile in terms of CTx and P1NP, and similar TBS changes after 1 year. Moreover, a similar panel of side effects was registered in both groups (9.2%, *p* = 0.862). The changes of glucose profile as reflected by fasting glucose and glycated hemoglobin A1c were not statistically significant upon 1 year of study in T2DM subjects [45].

Merugu et al. [49] included postmenopausal females with T2DM (N = 40; average age of 60.5 years old, mean diabetes duration of 9 ± 6.1 years) and non-diabetic controls (N = 40; average age of 57.5 years). At baseline, a similar BMD was identified in all three central sites (lumbar spine: *p* = 0.11; total hip: *p* = 0.50; femoral neck: *p* = 0.50). A higher proportion of patients with low TBS was found in the T2DM group *versus* controls, but this was not statistically significant (37.5% *versus* 25%, *p* = 0.23). All the Indian subjects included in this prospective cohort pilot study received a single injection of 4 mg zoledronic acid. After 1 year, the lumbar BMD increase was statistically significant in both groups (+3.6%, *p* = 0.01, respectively, +6.2%, *p* = 0.01), but the BMD gain was lower in the diabetic group. One explanation that was provided by the authors might be the blunt bone turnover markers profile at baseline (such as serum P1NP and β-CTX) [49].

## 4. Discussions

### 4.1. The Spectrum of “Sweet Bones”

Osteoporosis and diabetes represent conditions with an increased global prevalence, and both considerably contribute to the overall burden of disability, functional impairment, and reduced health-related quality of life, causing high costs and healthcare resource utilization, associated with a severe rate of morbidity and mortality. Any of the two diseases represent a major health issue, but when taken together, the picture is even more complicated [79,80,81].

Generally, a lower BMD is associated with a higher risk of fragility fracture [82]; however, the patients with T2DM might follow an apparently paradoxical combination of normal/increased BMD (as pointed out by the above cited studies from 2022–2023), but a higher risk of low-trauma/spontaneous fractures [16]. Additionally, T2DM has a higher risk than the general population to associate overweight and obesity, and this may also play a role in increased BMD [82,83]. However, it should be noted that these findings contrast with the data on BMD in type 1 diabetes mellitus (T1DM), in which BMD is typically lower than non-diabetic individuals, with T1DM being traditionally recognized as a cause of secondary osteoporosis with a 6.4- to 6.9-fold increase of fracture risk [84,85,86]. Concerning TBS, some authors showed that controlled T1DM is not correlated with a reduction of TBS [87], while others showed a decreased TBS in T1DM *versus* non-diabetic controls [88]; TBS was associated with prevalent fractures, but further studies are necessary to pinpoint the exact TBS cutoffs for fracture prediction in T1DM [89,90].

Overall, in terms of epidemiologic impact, 9 out of 10 individuals with diabetes have T2DM, but 1 out of 11 persons has any type of diabetes currently, both types being direct contributors to osteoporotic fractures [91,92]. Bone fragility in diabetes is caused by defects of mineralization and of trabecular microarchitecture in addition to other potential contributors to falling (as risk factor for fractures) such as diabetic sarcopenia, vitamin D deficiency, blood pressure and glycemic variations, impaired vision due to diabetic retinopathy, peripheral neuropathy, postural hypotension, foot amputation, etc. [93,94,95].

Glucose control in diabetic patients, as mentioned [40,41,46], might be directly associated with the skeleton health; however, the extent of quantifying long-term glycemic profile with respect to associated BMD, TBS, and fracture risk is rather difficult to be clearly presented based on current data so far (since there are no homogenous results in clinical trials). Clinically, patients with longer (≥10 years) disease duration and poor glycemic control have a higher risk of non-vertebral (including hip) fractures [96,97].

The main mechanism of T2DM-induced bone fragility involves advanced glycation end products (AGEs), an increased glycation of collagen leading to collagen-AGEs, which are related to the alteration of bone mineralization and quality [98,99]. Other contributors that may play a role in bone fragility are insulin resistance, adipose bone marrow alteration, inflammatory factors, and oxidative stress [100,101]. Recently, hypercholesterolemia and hypertriglyceridemia were found to be detrimental to bones, irrespective of mentioned glucose-related pathways [102]. Also, various methods of describing body composition features revealed different risk factors amid fat tissue distribution, not only direct body mass index influence, as already specified [47,48,50]. New growing evidence pointed out the potential involvement of microbiota (as part of the nutritional panel) in diabetic bone disease [103,104].

One of the most challenging aspects remains the inhibition of bone cells function and a decrease in bone turnover in diabetes [105,106], as previously mentioned with regard to TBS correlations in T2DM [40,41,48]. Bone formation marker osteocalcin has been found as a key regulator of metabolism–bone interplay [107,108]. Periostin, an emerging plasma marker of metabolism, might influence bone metabolism as well, with recent data suggesting its negative correlation with femoral neck and total hip BMD in T2DM and its positive association with osteocalcin levels [109] and potentially with non-vertebral fractures [110]. New data showed promising biomarkers such as increased galanin in the hypothalamus that was found in T2DM-associated osteoporosis underlying galanin control over insulin sensitivity and bone density [111].

Also, preptin, a peptide derivate from pro-IGF2 (insulin-like growth factor), seems to be involved not only in insulin resistance, but also in bone anabolism by stimulating osteoblasts activity and regulating osteocalcin secretion [112]. Notably, irisin, a muscle-produced hormone that was first identified a few years ago [113], was found to be correlated with the bone status in T2DM; namely, irisin was negatively correlated with *β*-CTX, being lower in osteoporotic subjects with newly detected T2DM *versus* those with normal DXA results according to one study published in 2022 [114] (Figure 2).

### 4.2. Integrating TBS to the Panel of Bone Status Assessment in T2DM

While DXA-BMD and associated T-score represents the gold standard for osteoporosis diagnosis regardless of the presence of diabetes, TBS is mandatory in T2DM menopausal females, but others adult categories might benefit, too. TBS rather than BMD is associated with vertebral fractures in T2DM [91,115,116,117]. FRAX (Fracture Risk Assessment) included T1DM as a secondary cause of osteoporosis, and not T2DM, while TBS-adjusted FRAX might prove useful in order to discriminate vertebral fractures [118]. As mentioned, bone turnover markers are beneficial in daily practice for T2DM patients, despite the fact that there are no standards to specifically address osteoporosis evaluation and some gaps are still a matter of research [119,120]. Vitamin D assays, particularly 25-hydroxyvitamin D, might prove hypovitaminosis D, which is correlated with metabolic components and bone status in T2DM [121]. The role of newly detected biomarkers such as irisin and others that involve signal transduction pathways of bone and metabolism, such as, for instance, those involving Wnt-beta catenin or myokines/adipokines panel, is yet to be identified [122] (Figure 3).

Several limits of the practical tools in T2DM-associated osteoporosis are represented by the usefulness of TBS in order to decide the medical intervention against osteoporosis and the expected TBS pattern under anti-osteoporotic drugs [123,124]. So far, the management of diabetic bone disease is similar to non-diabetic subjects (including bisphosphonates, denosumab, and teriparatide use) [125,126,127] in addition to glucose (and even lipids) lowering drugs to supplementary anti-fracture benefits [128]. As a dual approach, we mention the recently identified additional effects on the glucose profile of denosumab *versus* ibandronate in subjects with combined T2DM in terms of not only lowering fasting and postprandial glycaemia, but, also, of elevating GLP-1 (glucagon-like peptide) and decreasing dipeptidyl-peptidase (DPP-4) under RANKL (receptor activator of NF-κB ligand) inhibitor when compared to the mentioned intravenous bisphosphonate [129]. The exact mechanisms of interfering with glucose homeostasis are still unclear, but experimental studies on β-cells showed that the monoclonal antibody that blocks the RANK/RANKL pathway might protect the pancreatic cell against dysfunction and apoptosis induced by diabetes-induced inflammatory cytokines and reactive oxygen species [130]. Additionally, we mention further topics to explore such as the results of murine experiments showing linagliptin and metformin protective effects against bone loss [131] (one of the pathogenic traits being metformin-induced down regulation of RAGE-JAK2-STAT1 signal transduction pathway) [132]. Also, the metabolic components other than the anomalies of glucose profile, such as obesity, chronic steatosis, and abnormal lipid status might add a supplementary influence on bone damage that may be additional to insulin resistance, hyperinsulinemia, and increased blood glycemia together with the potential impairment of mineral metabolism [133,134,135].

Notably, the pitfall of normal/increased BMD according to central DXA exam might create an anti-osteoporotic treatment gap in T2DM adults and further randomized controlled trials are necessary to specifically address this issue [136,137]. Due to the epidemiologic impact and medical/social burdens that are associated with T2DM, it is mandatory to recognize and adequately approach the anomalies of bone status in T2DM including via TBS assessments.

Nevertheless, TBS use in T2DM has a first line indication to evaluate the skeleton health in terms of fragility fracture risk, as similarly seen in glucocorticoid-induced osteoporosis [43,138,139]. Other non-DM endocrine conditions might display a damage of bone microarchitecture, thus TBS found its way in daily practice in addition to other specific assessments in primary hyperparathyroidism, acromegaly, patients under chronic levothyroxine therapy for TSH (thyroid-stimulating hormone) suppression following thyroidectomy for differentiated thyroid cancer, etc. [140,141,142,143,144].

We are aware of the limitations regarding a narrative review with a 21-month research window on PubMed; however, we intended to provide a most recent update on such an effervescent topic with regard to one of the most fascinating cross-disciplinary fields of bone metabolic (type 2 diabetic) disorder. Our sample-based analysis (n = 11, N = 4653) confirmed the use of TBS in T2DM particularly in females (females/males ratio of 1.9), with ages varying between 35 and 91 years. With concern to the study design, apart from the transversal studies, two others were prospective, while another two were case-control. This early-post-pandemic analysis included studies of various sample sizes, such as: males and females (N of 245, 361, 511, and 2294), only women (N of 80, 96, 104, 243, 493, and 887), only men (N = 169) [40,41,42,43,44,45,46,47,48,49,50] (Figure 4).

## 5. Conclusions

Overall, this 21-month sample-based analysis confirmed prior data on BMD and TBS values in T2DM, while the issue of whether checking the fracture risk is mandatory in adults with uncontrolled T2DM remains to be proven or, on the other hand, a reduced TBS might function as a surrogate marker of complicated/uncontrolled T2DM. The interventional approach with bisphosphonates for treating T2DM-associated osteoporosis remains a standard one; as specified; further studies will pinpoint additive benefits on glucose status of anti-osteoporotic drugs or will confirm if certain glucose-lowering regimes are supplementarily beneficial for fracture risk reduction. These data showed once again that the patients with T2DM often have a lower TBS than those without diabetes or with normal glucose levels. Therefore, a decline in TBS may reflect an early stage of bone health impairment in diabetes. TBS derived from DXA images remains a useful non-invasive index of bone microarchitecture and can be an easily applicable tool for the diagnosis of bone fragility in patients with T2DM. Further studies are necessary to point out the TBS applications in order to decide the timing of anti-osteoporotic medication and long-term follow-up under such drugs.

## Figures and Tables

**Figure 1 jcm-12-07399-f001:**
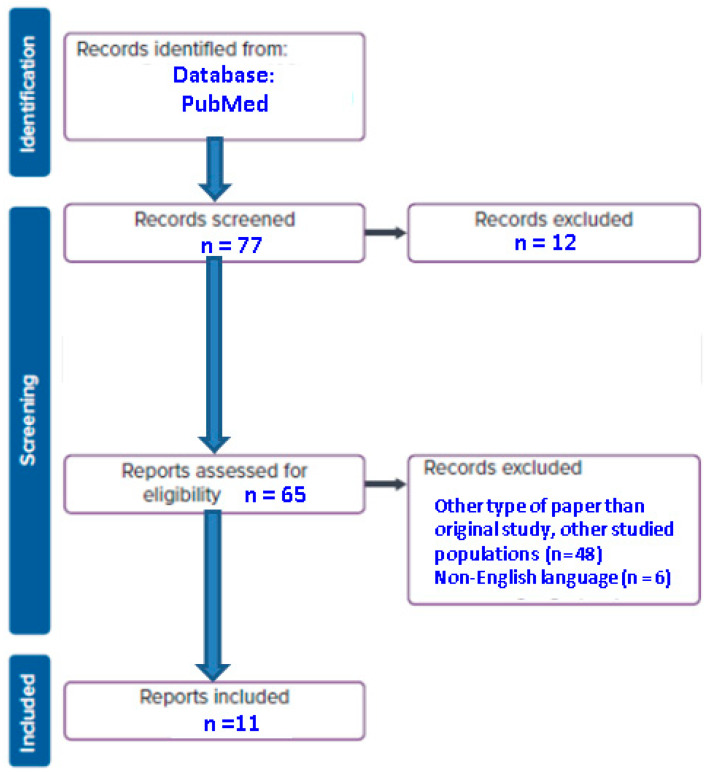
Flowchart diagram of included articles (n = number of articles).

**Figure 2 jcm-12-07399-f002:**
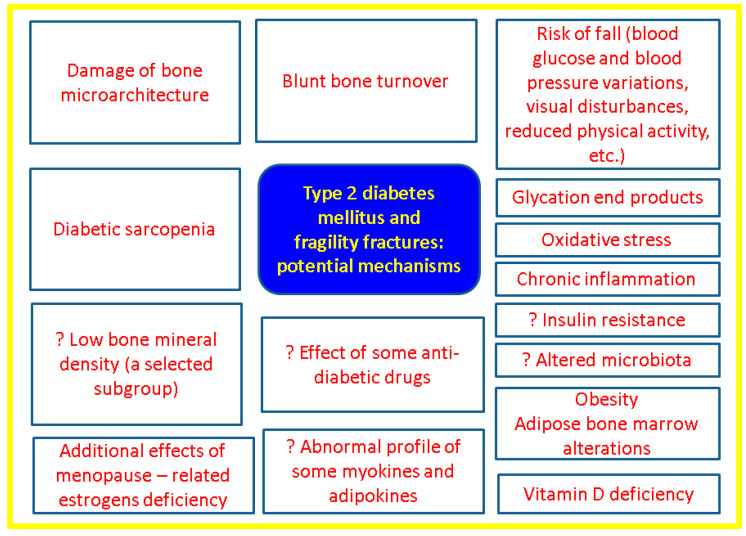
Overview of the main mechanisms of fragility fractures in T2DM [40,41,42,43,44,45,46,47,48,49,50,51,52,53,54,55,56,57,58,59,60,61,62,63,64,65,66,67,68,69,70,71,72,73,74,75,76,77,78,79,80,81,82,83,84,85,86,87,88,89,90,91,92,93,94,95,96,97,98,99,100,101,102,103,104,105,106,107,108,109,110,111,112,113,114].

**Figure 3 jcm-12-07399-f003:**
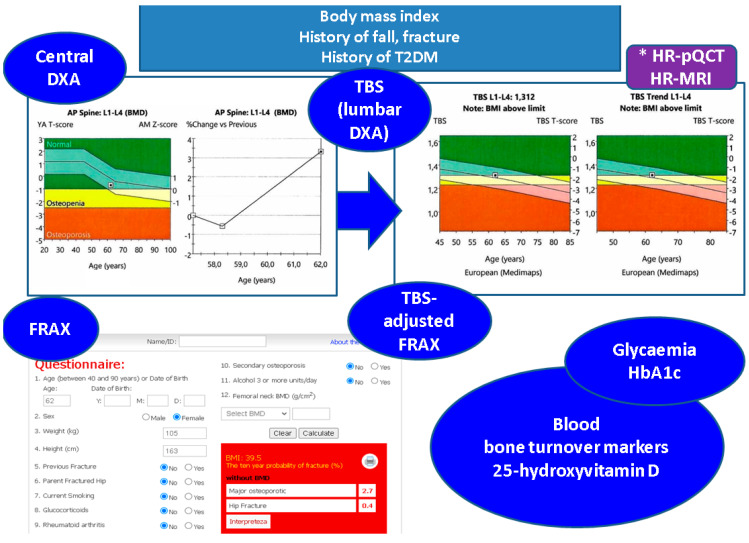
Practical tools to address skeleton heath in adults with T2DM: clinical evaluations includes body mass index while anamnesis highlights potential fracture risks such as history of falling prior diagnosis of osteoporosis, etc.; also, the records of diabetes history might be helpful in the matter of bone status; central DXA remains the gold standard for osteoporosis diagnosis (if any) and lumbar DXA assessment is the basis of TBS (alternatively, microarchitecture might be assessed via HR-pQCT or HR-MRI*), fracture risk assessment tool is applicable if studied population/patient is registered in the calculator (https://frax.shef.ac.uk/FRAX/tool.aspx Free access on 6 October 2023), including TBS-adjusted FRAX; blood tests are useful not only to reveal the glucose control, but also to provide bone turnover markers and vitamin D status (these DXA, TBS records, and FRAX result belong to a 62-year-old female associating obesity and T2DM with normal BMD and partially degraded bone microarchitecture as reflected by a TBS value of 1312) [40,41,42,43,44,45,46,47,48,49,50].

**Figure 4 jcm-12-07399-f004:**
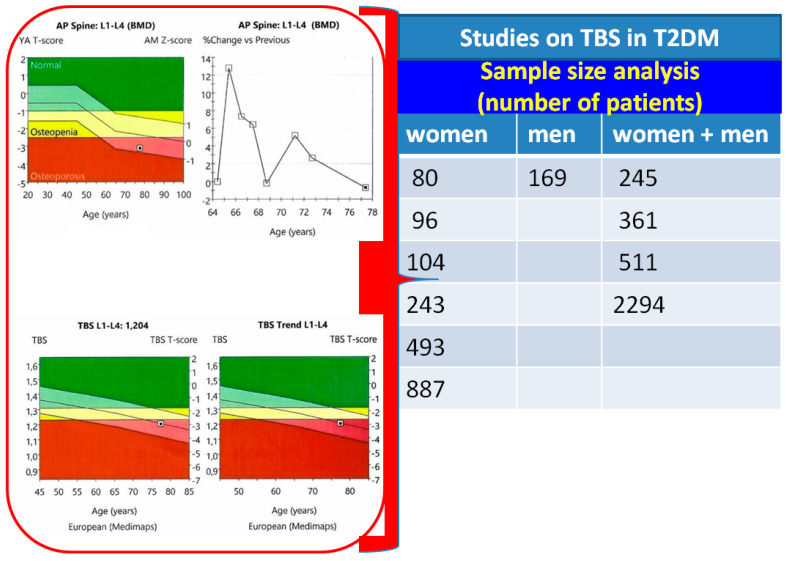
Sample size analysis on published studies [40,41,42,43,44,45,46,47,48,49,50] (the attached clinical vignette—red box—introduces a 77-year-old normal-weighted female with T2DM who has a long history of bisphosphonates therapy for osteoporosis; TBS of 1204 shows a degraded microarchitecture).

**Table 1 jcm-12-07399-t001:** Most important findings according to the original studies we identified based on mentioned methods with regard to TBS and T2DM [40,41,42,43,44,45,46,47,48,49,50].

First Author Year of Publication Reference Number	Study Design Number of Patients Sex ratio (F/M) Age (Years)	Core TBS Findings at Baseline
Maamar El Asri, 2022 [40]	Cross-sectional study N = 361 with T2DM F/M = 184/177 Mean age: 63.8 y (47–91 y) N1 = 92 with microvascular disease N2 = 269 without microvascular disease	N1: TBS = 1.235 ± 0.1 N2: TBS = 1.287 ± 0.1 Patients with diabetic microvascular disease had a statistically significant lower TBS than the patients without microvascular disease: N1 vs. N2, *p* = 0.034
Ballato, 2022 [41]	Cross-sectional study N = 169 males Mean age: 51.4 ± 7.5 y (35–65 y) N1 = 91 without T2DM N2 = 26 with HbA1c ≤ 7% N3 = 52 with HbA1c > 7%	N1: TBS = 1.26 ± 0.15 N2: TBS = 1.22 ± 0.12 N3: TBS = 1.21 ± 0.15 No significant differences in TBS among the patients with good vs. poor glycemic control: N1 vs. N2 vs. N3, *p* = 0.28
Fazullina, 2022 [42]	Cross-sectional study N = 96 postmenopausal females with T2DM and normal BMD Mean age: 64 y (50–75 y) N1 = 53 with TBS > 1.31 N2 = 43 with TBS ≤ 1.31	N1: TBS = 1.465 (1.39–1.514) N2: TBS = 1.206 (1.127–1.271) Postmenopausal females with T2DM and normal BMD may have impaired bone microarchitecture; a decrease in TBS (≤1.31) was observed in 44.8% of study subjects Prevalence of fractures was higher in N2 group than N1 (32.6% vs. 11.3%, *p* = 0.02)
Gharibzadeh, 2022 [43]	Cross-sectional study N = 2294 F/M = 1182/1112 Mean age: 69.3 ± 6.3 y Females N1 = 412 with T2DM N2 = 770 non-diabetic controls Males N1′ = 314 with T2DM N2′ = 798 non-diabetic controls	Females N1: TBS = 1.23 ± 0.09 N2: TBS = 1.24 ± 0.08 Males N1′: TBS = 1.36 ± 0.09 N2′: TBS = 1.35 ± 0.08 T2DM had a significant effect only in men’s TBS (*p* = 0.03)
Haeri, 2022 [44]	Cohort study N = 511 F/M = 433/78 Mean age females: 80.6 ± 8.0 y Mean age males: 82.4 ± 8.3 y Females N1 = 105 with T2DM N2 = 328 non-diabetic controls Males N1 = 18 with T2DM N2 = 60 non-diabetic controls	Females N1: TBS = 1.211 ± 0.172 N2: TBS = 1.266 ± 0.136 Males N1: TBS = 1.255 ± 0.189 N2: TBS = 1.268 ± 0.132 Diabetic females compared with nondiabetics had lower spine TBS (*p* = 0.0299), but no differences between groups in males (*p* = 0.7935)
Kim, 2022 [45]	Prospective study N = 104 postmenopausal females N1 = 49 with T2DM Mean age: 73 y (67–79 y) N2 = 55 non-diabetic controls Mean age: 66 y (63–73 y)	N1: TBS = 1.289 ± 0.076 N2: TBS = 1.300 ± 0.058 At baseline, there was no difference in TBS between groups (*p* = 0.294)
Palomo, 2022 [46]	Cross-sectional study N = 493 females Mean age: 71.8 y N1 = 116 with HbA1c ≥ 6.5% N2 = 217 with HbA1c 5.7–6.4% N3 = 160 with HbA1c ≤ 5.6%	N1: TBS = 1.280 ± 0.109 N2: TBS = 1.299 ± 0.093 N3: TBS = 1.314 ± 0.104 TBS was lower in patients with higher HbA1c (*p* = 0.020)
Dule, 2023 [47]	Observational case-control study N = 243 N1 = 126 females with T2DM Mean age: 62.96 ± 6.73 y N2 = 117 non-diabetic controls Mean age: 61.91 ± 5.8 y	N1: TBS = 1.180 ± 0.112 N2: TBS = 1.209 ± 0.120 T2DM was associated with low TBS (OR = 2.47, 95% CI: 1.19–5.16, *p* = 0.016) in a regression model adjusted for age, menopausal status and BMI
Ubago-Guisado, 2023 [48]	Case-control study N = 245 N1 = 111 with T2DM F/M = 48/63 Mean age: 65.4 ± 7.6 y N2 = 134 non-diabetic controls F/M = 65/69 Mean age: 64.7 ± 8.6 y	N1: TBS = 1.074 ± 0.187 N2: TBS = 1.291 ± 0.110 TBS was lower in the T2DM group compared to the controls (*p* < 0.001)
Merugu, 2023 [49]	Prospective cohort study N = 80 N1 = 40 females with T2DM Mean age: 60.5 y (57.2–65 y) N2 = 40 non-diabetic controls Mean age: 57.5 y (53–64.7 y)	N1: TBS = 1.24 ± 0.07 N2: TBS = 1.26 ± 0.08 At baseline, TBS was similar between groups (*p* = 0.25)
Naseri, 2023 [50]	Cross-sectional study N = 887 N1 = 348 postmenopausal females Mean age: 55.13 ± 6.61 y N2 = 539 non-diabetic controls Mean age: 55.13 ± 6.61 y	N1: TBS = 1.280 ± 0.111 N2: TBS = 1.343 ± 0.101 TBS was statistically significantly lower in diabetic subjects than in non-diabetic controls (*p* < 0.001)

Abbreviations: BMD = bone mineral density, BMI = body mass index, F = female, HbA1c = glycated hemoglobin, M = male, N = number of patients, OR = odds ratio, TBS = trabecular bone score, T2DM = type 2 diabetes mellitus, vs. = versus, y = year.

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
