# Peer review of "Trabecular Bone Score (TBS) in Individuals with Type 2 Diabetes Mellitus: An Updated Review"

_jcm, 2023, doi:10.3390/jcm12237399_

Round 1

Reviewer 1 Report

Comments and Suggestions for Authors

The manuscript by Trandafir et al., is well written and provides an update on the potential clinical significance of TBS in type 2 diabetes. There are minor language edits required and I will suggest that the title be modified to reflect that this is an update on this topic; something in the line of "an updated review on Trabecular Bone Score (TBS) in individuals with Type 2 Diabetes Mellitus" or "Trabecular Bone Score (TBS) in individuals with Type 2 Diabetes Mellitus: an updated review"

Comments on the Quality of English Language

There are minor english language edits required

Author Response

Response to Review 1 Comments

Dear Reviewer,

Thank you very much for your time and your effort to review our manuscript.

We are very grateful for providing your valuable feedback on the article.

Here is our response and related amendment that has been made in the manuscript according to your review (marked in yellow color).

The manuscript by Trandafir et al. is well written and provides an update on the potential clinical significance of TBS in type 2 diabetes.

Thank you very much. We really appreciate it!

There are minor language edits required and I will suggest that the title be modified to reflect that this is an update on this topic; something in the line of "an updated review on Trabecular Bone Score (TBS) in individuals with Type 2 Diabetes Mellitus" or "Trabecular Bone Score (TBS) in individuals with Type 2 Diabetes Mellitus: an updated review".

Thank you very much. We followed your recommendation and changed the title into "Trabecular Bone Score (TBS) in individuals with Type 2 Diabetes Mellitus: an updated review". Thank you

There are minor English language edits required

Thank you very much. We re-checked the text. Thank you.

Thank you very much.

Reviewer 2 Report

Comments and Suggestions for Authors

A brief review on Trabecular Bone Score (TBS) in Type 2 Diabetes Mellitus

Title

-          To make the study clearer, please mention the study design in the study title.

Abstract

-          The research gap of the study was presented in the abstract section. Please improve the novelty of the study in the abstract section.

-          Provide the study setting.

-          Provide the critical value of the statistical analysis result, such as the p-value.

Introduction

-          Please emphasise the highlighted topic in each paragraph by making a simple conclusion in each paragraph.

-          Please increase the scientific background of why the study topic must be conducted, particularly the urgency of bone fragility among T2DM patients.

-          Please promote the novelty of the study. Please state the novelty that would be provided in this study regarding bone fragility among T2DM patients.

-          Please promote the primary variable of this study and increase the discussion of the main variable in the introduction section.

-          Please improve the study aims to be more specific and detailed.

Method

-          Why did not the author utilize the method of meta-analysis? so the study result would be more valid due to the nature of the meta-analysis procedure.

-          Why does the author only select the study within the last year?

-          What is the method to analyze the study result?

Result

-          Since the introduction did not provide information on the main variable, thus the study results just come with a new perspective without the scientific reason. 

Discussion

-          Besides the main result, please raise the scientific problem about the variables and promote this study’s result as the strategy to solve its scientific problem.

-          How about the study generalizability issue?

Author Response

Response to Review 2 Comments

Dear Reviewer,

Thank you very much for your time and your effort to review our manuscript.

We are very grateful for your insightful comments and observations, also, for providing your valuable feedback on the article.

Here is a point-by-point response and related amendments that have been made in the manuscript according to your review (marked in yellow color).

Title - To make the study clearer, please mention the study design in the study title.

Thank you very much. We followed the recommendation and changed the title to "Trabecular Bone Score (TBS) in individuals with Type 2 Diabetes Mellitus: an updated review" since this is a review as study design.

The Abstract also highlights the fact that this paper is a narrative review “The core of this narrative review….”

At Methods section we also specified this type of approach “we introduced our analysis according to a narrative review.”

 Thank you

Abstract-The research gap of the study was presented in the abstract section. Please improve the novelty of the study in the abstract section.

Thank you very much. The following 3 statements conclude the novelty of this review since the paper is a review:

….”patients with T2DM often have a lower TBS than those without diabetes or with normal glucose levels”.

….”decline in TBS may reflect an early stage of bone health impairment in diabetes”.

..”TBS remains a useful non-invasive index of bone microarchitecture and can be an easy applicable tool for assessing bone fragility in T2DM.”

Thank you.

Abstract - Provide the study setting.

Thank you very much. The methods of this review include:

Study design = narrative review (line 30)

Types of research papers = original, English published, full length studies (line 31)

Database of research: PubMed (line 31)

Time frame: latest 21 months (line 37)

Thank you.

Abstract-Provide the critical value of the statistical analysis result, such as the p-value.

Thank you very much. We followed your recommendation. Among the many p-values we mentioned across the paper according to the review, we selected an intersting study results for the Abstract and introduced them as followings:

“One control study on 4 mg zoledronic acid showed after 1 year a statistically significant increase of lumbar BMD in both diabetic and non-diabetic group (+3.6%, p = 0.01, respectively, +6.2%, p = 0.01)”.

Thank you

Introduction-Please emphasise the highlighted topic in each paragraph by making a simple conclusion in each paragraph.

Thank you very much. At Introduction section we displayed 4 main paragraphs and each of them starts with a highlighted topic as you recommended, namely:

The first paragraph – Bone fragility as a new complication of types 2 diabetes mellitus as it has been detected during 10 years.

“Bone fragility is a complication of type 2 diabetes mellitus (T2DM) that has been identified amid latest decade…”

The second paragraph – The importance of taking into consideration the topic of type 2 diabetes mellitus is due to its massive epidemiologic impact.

“T2DM, a chronic metabolic condition, has a massive public health burden...”

The third paragraph – The importance of taking into consideration DXA assessment as gold standard evaluation for osteoporosis that, however, it is not as useful as expected in patients diagnosed with type 2 diabetes mellitus.

 “BMD and associated T-score according to central DXA are recommended for screening in order to evaluate the fracture risk; however, BMD does not fully reflect….”

The fourth paragraph – The paradox of having normal or high bone mineral density in subjects confirmed with type 2 diabetes mellitus.

 “T2DM is characterized by normal or high BMD, but an increased risk of fragility fractures, an apparent paradox”…

Thank you.

Introduction-Please increase the scientific background of why the study topic must be conducted, particularly the urgency of bone fragility among T2DM patients.

Thank you very much. According to your recommendation, we specified the importance of treating as urgency the bone fragility status among patients diagnosed with type 2 diabetes mellitus.

“Overall, the affected bone microarchitecture is recognized as a major factor contributing to the risk of fracture in T2DM, diabetic bone disease representing a major co-morbidity as part of an already complicated panel, thus the importance of addressing such topic in modern era [17-19], noting once again the importance of understanding and studding the bone fragility in T2DM due to increased burden in terms of morbidity and even mortality especially when it comes to the hip fractures that affects a large number of population.”

 Thank you

Introduction- Please promote the novelty of the study. Please state the novelty that would be provided in this study regarding bone fragility among T2DM patients.

Thank you very much. The novelty comes from the statement related to the fact that only amid latest decade bone fragility has been recognized as a complication of an otherwise challenging panel of complications in type 2 diabetes (first paragraph).

The second aspect comes from the modern era that encourages the use of TBS as a new tool for practitioners (not only in the field of osteoporosis) to address bone fragility.

Both aspects have been highlighted at Introduction section. The third aspect of novelty is reflected by the time frame of research, namely the latest 21 months. Thank you

Introduction-Please promote the primary variable of this study and increase the discussion of the main variable in the introduction section.

Thank you very much. The primary variable is TBS and we highlighted the following main data at Introduction:

  1. Trabecular bone score (TBS) is obtained from DXA scans of the lumbar spine
  2. TBS provides information independently of BMD
  3. TBS is indirectly calculated from using experimental variograms
  4. TBS indicates a reduced number of trabeculae and less connectivity
  5. TBS is the most widely used instrument to study bone microarchitecture in diabetes
  6. Lower values of TBS were associated with an increased risk of fragility fracture
  7. BMD paradox in type 2 diabetes versus TBS

Thank you

Introduction-Please improve the study aims to be more specific and detailed.

Thank you very much. We expanded and improved the specifications amid the aim of this work according to your recommendations, as followings:

“We followed several aspects such as TBS values in relationship with DXA-BMD, the reflection of glucose profile into TBS, associated anomalies of metabolism that might interfere with bone status in these patients, the correlation between TBS and bone turnover as reflected by blood/urinary biomarkers assays and the specific intervention against osteoporosis in T2DM with potential effects on TBS improvement.”

Thank you

Method-Why did not the author utilize the method of meta-analysis? so the study result would be more valid due to the nature of the meta-analysis procedure.

Thank you very much. Indeed, this is a narrative review, not a systematic one or meta-analysis. Due to the heterogeneity of the spectrum with respect to TBS-T2DM, we choose to introduce the data as a narrative review since various levels of statistical evidence are identified in the mentioned papers (meaning different types of studies with different studied parameters, cohorts, data, results, assessments, and outcomes).

On the other hand, a meta-analysis pinpoints a specific critical assessment which in the matter of TBS-T2DM seems less useful so far from a practical, multidisciplinary perspective; for instance, only two studies addressed the issue of the specific anti-osteoporotic medication in the mentioned studied population diagnosed with type 2 diabetes mellitus.

However, this type of review is a well-recognized, standard, traditional approach. This allowed us to examine and evaluate the scientific panel on this specific topic in a useful way for various practitioners. 

We mentioned at Discussion the aspect of this research/paper approach.

“We are aware of the limitations regarding a narrative review”…

Thank you

Method-Why does the author only select the study within the last year?

Thank you very much. This is a 21-month analysis on the most recent data that is why the paper is called now “updated review”.

The selected time frame is between 1st January 2022 and 30th September 2023, as we specified at Methods section, in order to provide the latest researches in this particular matter.  Thank you

Method-What is the method to analyze the study result? 

Thank you very much. The method of analysing the cited studies across our review stands firstly for selecting the papers (according to the mentioned criteria such as study design, time frame, PubMed access, English language). Further on, the specific inclusion and exclusion criteria have been specified.

“Based on the hypothesis that diabetes is associated with abnormal bone microarchitecture, T2DM correlates with a lower TBS values when compare to non-diabetic subjects. We therefore set out to search the literature for studies that have examined TBS in patients with T2DM published in PubMed (full-length articles, published in English, between 1st January 2022 and 30th September 2023) and we introduced our analysis according to a narrative review. The keywords used for the search were “trabecular bone score” (alternatively, “TBS”) and “diabetes”. Inclusion criteria are represented by original studies of any study design conducted in adults (females and males) diagnosed with T2DM. We excluded: review papers, case reports and case series, editorials, letters to the editor, editorials, conference abstracts/reports, animal/experimental studies, data on type 1 diabetes mellitus (T1DM) and other secondary causes of diabetes, as well prediabetes/impaired glucose tolerance, and pediatric data. This 21-month sample – based analysis followed several micro-sections such as DXA – based BMD and TBS aspects, TBS values in relationship with glucose control, bone turnover markers and interventional approach for diabetes – associated osteoporosis.”

The flowchart of the study is displayed in Figure 1.

Finally, the data we found were mentioned across 6 main sections at Results (from 3.1. to 3.6.) and associated tables. All the relevant statistical results have been displayed and discussed. Thank you

Result-Since the introduction did not provide information on the main variable, thus the study results just come with a new perspective without the scientific reason. 

Thank you very much. The facts regarding TBS in Introduction have already been explained. The main variable in this review is TBS in type 2 diabetic patients. Each of the subsections from Results section follows the values of this parameter amid different outcomes such as DXA-BMD, osteoporosis therapy, correlations with bone turnover makers, etc. Not only that we commented all these aspects and even more, but we highlighted them in the table, for instance, thus respecting your recommendation (we provide the extract below):

Table 1

Core TBS findings at baseline

N1: TBS = 1.235 ± 0.1

N2: TBS = 1.287 ± 0.1

Patients with diabetic microvascular disease had a statistically significant lower TBS than the patients without microvascular disease: N1 vs. N2, p = 0.034

N1: TBS = 1.26 ± 0.15

N2: TBS = 1.22 ± 0.12

N3: TBS = 1.21 ± 0.15

No significant differences in TBS among the patients with good vs. poor glycemic control: N1 vs. N2 vs. N3,  p = 0.28

N1: TBS = 1.465 (1.39-1.514)

N2: TBS = 1.206 (1.127-1.271)

Postmenopausal females with T2DM and normal BMD may have impaired bone microarchitecture, a decrease in TBS (≤1.31) was observed in 44.8% of study subjects

Prevalence of fractures was higher in N2 group than N1 (32.6% vs. 11.3%, p = 0.02)

Females

N1: TBS = 1.23 ± 0.09

N2: TBS = 1.24 ± 0.08

Males

N1’:  TBS = 1.36 ± 0.09

N2’: TBS = 1.35 ± 0.08

T2DM had a significant effect only in men’s TBS (p = 0.03)

Females

N1: TBS = 1.211 ± 0.172

N2: TBS = 1.266 ± 0.136

Males

N1: TBS = 1.255 ± 0.189

N2: TBS = 1.268 ± 0.132

Diabetic females compared with nondiabetics had lower spine TBS (p = 0.0299), but no differences between groups in males (p = 0.7935)

N1: TBS = 1.289  ±  0.076

N2: TBS = 1.300 ± 0.058

At baseline, there was no difference in TBS between groups (p = 0.294)

N1: TBS = 1.280 ± 0.109

N2: TBS = 1.299 ± 0.093

N3: TBS = 1.314 ± 0.104

TBS was lower in patients with higher HbA1c (p = 0.020)

N1: TBS = 1.180 ± 0.112

N2: TBS = 1.209 ± 0.120

T2DM was associated with low TBS (OR = 2.47, 95% CI: 1.19–5.16, p = 0.016) in a regression model adjusted for age, menopausal status and BMI

N1: TBS = 1.074 ± 0.187

N2: TBS = 1.291 ± 0.110

TBS was lower in the T2DM group compared to the controls (p < 0.001)

N1: TBS = 1.24 ± 0.07

N2: TBS = 1.26 ± 0.08

At baseline, TBS was similar between groups (p = 0.25)

N1: TBS = 1.280 ± 0.111

N2: TBS = 1.343 ± 0.101

TBS was statistically significant lower in diabetic subjects than in non-diabetic controls (p < 0.001)

Thank you

Discussion-Besides the main result, please raise the scientific problem about the variables and promote this study’s result as the strategy to solve its scientific problem.

Thank you very much. Based on your recommendation, we highlighted the followings at Discussion section:

“Notably, the pitfall of normal/increased BMD according to central DXA exam might create an anti-osteoporotic treatment gap in T2DM adults and further randomized controlled trials are necessary to specifically address this issue... Due to the epidemiologic impact and medical/social burden that are associated with T2DM it is mandatory to recognize and adequately approach the anomalies of bone status in T2DM including via TBS assessments.”

Thank you

Discussion-How about the study generalizability issue?

Thank you very much. Since this is a narrative review, this comes as general as possible underlying the specific profile of all the mentioned original studies that have been addressed within the main text and Table 1. Thank you

Round 2

Reviewer 2 Report

Comments and Suggestions for Authors

A brief review on Trabecular Bone Score (TBS) in Type 2 Diabetes Mellitus

Abstract

-       The research gap of the study was presented in the abstract section. Please improve the novelty of the study in the abstract section.

Introduction

-       Please emphasise the highlighted topic in each paragraph by making a simple conclusion in each paragraph.

-       Please increase the scientific background of why the study topic must be conducted, particularly the urgency of bone fragility among T2DM patients.

-       Please promote the novelty of the study. Please state the novelty that would be provided in this study regarding bone fragility among T2DM patients.

Author Response

Response to Review 2 Comments – second round

Dear Reviewer,

Thank you very much for your time and your effort to review our manuscript.

We are very grateful for providing your valuable feedback on the article.

Here is our response and related amendment that has been made in the manuscript according to your review (marked in green color).

Abstract: The research gap of the study was presented in the abstract section. Please improve the novelty of the study in the abstract section.

Thank you very much. We corrected upon your recommendation:

“The novelty of this literature research: these insights showed once again that the patients with T2DM often have a lower TBS than those without diabetes or with normal glucose levels. Therefore, the decline in TBS may reflect an early stage of bone health impairment inT2DM. The novelty of the TBS as a handy, non-invasive method that proved to be an index of bone microarchitecture confirms its practicality as an easy applicable tool for assessing bone fragility in T2DM.“

Thank you

…………………………

Introduction: Please emphasise the highlighted topic in each paragraph by making a simple conclusion in each paragraph.

Thank you very much. These are the simple conclusions of each paragraph we introduced based on your recommendation:

“Thus, the impairment of bone status in T2DM is mandatory to be assessed.”

“Therefore, the increasing number of diabetic patients requires a detailed evaluation, including fracture risk.”

“Consequently, TBS remains an important tool to be analyzed in relationship with T2DM.”

“…hence the importance of addressing such topic in modern era.”

Thank you

………………….

Introduction: Please increase the scientific background of why the study topic must be conducted, particularly the urgency of bone fragility among T2DM patients.

Thank you very much. Based on your recommendation, we introduced a new section at Introduction chapter called “The scientific background of why the study topic must be conducted:”

Also, we added the following scientific data:

“The scientific rational of detecting and improving the strategy of addressing the osteoporotic fractures in general population(and in type 2 diabetic individuals in particular)  comes from a major burden that may be reflected by impressive data: the mortality within the first year following a hip fracture may arise to 25% while disability and dysfunctionality might affect up to 60% of affected persons following this type of fracture; the projections of osteoporotic fractures as increasing number of patients involve both females and males; a call for action has been released by World Health Organization to address the issue of fragility fractures and osteoporosis detection due to associated burden; remaining life time probability of suffering from a fracture is 1 out of 3 females and 1 out of 5 men aged over 50 in Europe, respectively, 1 out of 2, and 1 out of 4 in US; the number and costs of hospitalizations due to fractures surpasses those related to cardiovascular events; the panel of osteoporosis - associated disability represents a heterogeneous spectrum that is also reflected by the aggravation of other co-morbidities, increased costs, and an overall impaired quality of life. On the other hand, only 2 out of 10 patients with osteoporosis are recognized with this diagnosis, and only 3 out of 10 subjects with a fracture who should receive specific anti-osteoporotic medication are actually treated for this condition in order to reduce the increased fracture risk, while almost half of the patients that start this medication will no longer be compliant after the first year since drug initiation [20-26].

Importantly, the burden of T2DM comes not only by its alarming globally increasing incidence, including in younger people and low-income countries (affecting from 6% to 22% from entire population at different age segments), but, also, by the associated morbidity and mortality; for instance, it hinders the ninth place among the most frequent causes of death, and the seventh place among the conditions with the most impactful quality of life in terms of DALYs (disability-adjusted life year), equality affecting females and males, the costs per capita of a diabetic care are 3 to 9 times (in cases of complications) higher than mean healthcare expenditure per capita in non-diabetic population [27-29].

While modern era is marked by aging population [30], both osteoporosis and T2DM might come as a mandatory issues to be addressed from all multidisciplinary perspectives from primary to tertiary care, from medical to social approaches [31-39]. Noting these important mentioned aspects, the assessment of bone fragility among T2DM comes as urgency.”

Moreover, we added the references backup of the mentioned scientific background by adding 20 more references and an adjustment to all the other references number until 124, not 144.

  1. https://www.osteoporosis.foundation/policy-makers/burden-osteoporosis
  2. Shoback D, Rosen CJ, Black DM, Cheung AM, Murad MH, Eastell R. Pharmacological Management of Osteoporosis in Postmenopausal Women: An Endocrine Society Guideline Update. J Clin Endocrinol Metab. 2020;105(3):dgaa048. doi:10.1210/clinem/dgaa048.
  3. Eastell R, Rosen CJ, Black DM, Cheung AM, Murad MH, Shoback D. Pharmacological Management of Osteoporosis in Postmenopausal Women: An Endocrine Society* Clinical Practice Guideline. J Clin Endocrinol Metab. 2019;104(5):1595-1622. doi:10.1210/jc.2019-00221.
  4. Moayyeri A, Warden J, Han S, Suh HS, Pinedo-Villanueva R, Harvey NC, Curtis JR, Silverman S, Multani JK, Yeh EJ. Estimating the economic burden of osteoporotic fractures in a multinational study: a real-world data perspective. Osteoporos Int. 2023. doi:10.1007/s00198-023-06895-4. 
  5. Yeh EJ, Rajkovic-Hooley O, Silvey M, Ambler WS, Milligan G, Pinedo-Villanueva R, Harvey NC, Moayyeri A. Impact of fragility fractures on activities of daily living and productivity in community-dwelling women: a multi-national study. Osteoporos Int. 2023;34(10):1751-1762. doi:10.1007/s00198-023-06822-7.
  6. Harvey NC, Poole KE, Ralston SH, McCloskey EV, Sangan CB, Wiggins L, Jones C, Gittoes N, Compston J; ROS Osteoporosis and Bone Research Academy Investigators. Towards a cure for osteoporosis: the UK Royal Osteoporosis Society (ROS) Osteoporosis Research Roadmap. Arch Osteoporos. 2022;17(1):12. doi:10.1007/s11657-021-01049-7.
  7. GBD 2021 Diabetes Collaborators.
    Global, regional, and national burden of diabetes from 1990 to 2021, with projections of prevalence to 2050: a systematic analysis for the Global Burden of Disease Study 2021. 2023;402(10397):203-234. doi:10.1016/S0140-6736(23)01301-6.
  8. Khan MAB, Hashim MJ, King JK, Govender RD, Mustafa H, Al Kaabi J. Epidemiology of Type 2 Diabetes - Global Burden of Disease and Forecasted Trends. J Epidemiol Glob Health. 2020;10(1):107-111. doi:10.2991/jegh.k.191028.001.
  9. Leslie RD, Ma RCW, Franks PW, Nadeau KJ, Pearson ER, Redondo MJ. Understanding diabetes heterogeneity: key steps towards precision medicine in diabetes. Lancet Diabetes Endocrinol. 2023;11(11):848-860. doi:10.1016/S2213-8587(23)00159-6. 
  10. Zhang K, Ma Y, Luo Y, Song Y, Xiong G, Ma Y, Sun X, Kan C. Metabolic diseases and healthy aging: identifying environmental and behavioral risk factors and promoting public health. Front Public Health. 2023;11:1253506. doi:10.3389/fpubh.2023.1253506.
  11. Dyrek N, Wikarek A, Niemiec M, Kocełak P. Selected musculoskeletal disorders in patients with thyroid dysfunction, diabetes, and obesity. 2023;61(4):305-317. doi:10.5114/reum/170312. 
  12. Meier C, Eastell R, Pierroz DD, Lane NE, Al-Daghri N, Suzuki A, Napoli N, Mithal A, Chakhtoura M, Fuleihan GE, Ferrari S. Biochemical Markers of Bone Fragility in Patients With Diabetes.J Clin Endocrinol Metab. 2023:dgad255. doi: 10.1210/clinem/dgad255.
  13. Meier C, Eastell R, Pierroz DD, Lane NE, Al-Daghri N, Suzuki A, Napoli N, Mithal A, Chakhtoura M, El-Hajj Fuleihan G, Ferrari S. Biochemical Markers of Bone Fragility in Patients with Diabetes. A Narrative Review by the IOF and the ECTS. J Clin Endocrinol Metab. 2023:dgad255. doi:10.1210/clinem/dgad255.
  14. Rubin MR. Bone cells and bone turnover in diabetes mellitus. Curr Osteoporos Rep. 2015 Jun;13(3):186-91. doi: 10.1007/s11914-015-0265-0.
  15. Rubin MR. Skeletal fragility in diabetes. Ann N Y Acad Sci. 2017 Aug;1402(1):18-30. doi: 10.1111/nyas.13463.
  16. Leslie WD, Rubin MR, Schwartz AV, Kanis JA. Type 2 diabetes and bone. J Bone Miner Res. 2012 Nov;27(11):2231-7. doi: 10.1002/jbmr.1759.
  17. Khosla S, Samakkarnthai P, Monroe DG, Farr JN. Update on the pathogenesis and treatment of skeletal fragility in type 2 diabetes mellitus. Nat Rev Endocrinol. 2021 Nov;17(11):685-697. doi:10.1038/s41574-021-00555-5.
  18. Cortet B, Lucas S, Legroux-Gerot I, Penel G, Chauveau C, Paccou J. Bone disorders associated with diabetes mellitus and its treatments. Joint Bone Spine. 2019;86(3):315-320. doi:10.1016/j.jbspin.2018.08.002.
  19. Lekkala S, Taylor EA, Hunt HB, Donnelly E. Effects of Diabetes on Bone Material Properties. Curr Osteoporos Rep. 2019;17(6):455-464. doi:10.1007/s11914-019-00538-6.
  20. Bonaccorsi G, Messina C, Cervellati C, Maietti E, Medini M, Rossini M, Massari L, Greco P. Fracture risk assessment in postmenopausal women with diabetes: comparison between DeFRA and FRAX tools. Gynecol Endocrinol. 2018;34(5):404-408. doi:10.1080/09513590.2017.1407308.

Thank you

………………….

Introduction: Please promote the novelty of the study. Please state the novelty that would be provided in this study regarding bone fragility among T2DM patients.

Thank you very much. Based on your recommendation, we specified the followings:

“The novelty of such literature – based update should provide new perspectives of the means to assess the bone fragility status among T2DM patients in relationship with TBS that is placed in the core of these assessments; its correlation with traditional bone evaluation based on DXA, particularly BMD and BMD – derivate T-score, with blood and/or urinary bone turnover markers, and, also, with the panel of T2DM such as glucose profile (fasting glucose, postprandial glycaemia, and glycated hemoglobin A1c )and specific therapy against diabetes.”

Thank you very much. 
